# 5-Fluorouracil treatment induces characteristic T>G mutations in human cancer

Sharon Christensen [1], Bastiaan Van der Roest [1], Nicolle Besselink[1], Roel Janssen[1], Sander Boymans [1], John W.M. Martens [2,3], Marie-Laure Yaspo [4], Peter Priestley [5], Ewart Kuijk [1], Edwin Cuppen [1,3,6]* & Arne Van Hoeck [1]

5-Fluorouracil (5-FU) is a chemotherapeutic drug commonly used for the treatment of solid cancers. It is proposed that 5-FU interferes with nucleotide synthesis and incorporates into DNA, which may have a mutational impact on both surviving tumor and healthy cells. Here, we treat intestinal organoids with 5-FU and find a highly characteristic mutational pattern that is dominated by T>G substitutions in a CTT context. Tumor whole genome sequencing data confirms that this signature is also identified in vivo in colorectal and breast cancer patients who have received 5-FU treatment. Taken together, our results demonstrate that 5-FU is mutagenic and may drive tumor evolution and increase the risk of secondary malignancies. Furthermore, the identified signature shows a strong resemblance to COSMIC signature 17, the hallmark signature of treatment-naive esophageal and gastric tumors, which indicates that distinct endogenous and exogenous triggers can converge onto highly similar mutational signatures.

[1] Center for Molecular Medicine and Oncode Institute, University Medical Center Utrecht, Universiteitsweg 100, 3584 CG Utrecht, The Netherlands. [2] Department of Medical Oncology, Erasmus MC Cancer institute, Erasmus University Medical Center, Doctor Molewaterplein 40, 3015 GD Rotterdam, The Netherlands. [3] Center for Personalized Cancer Treatment, Rotterdam, The Netherlands. [4] Max Planck Institute for Molecular Genetics, Ihnestraße 63, 14195 Berlin, Germany. [5] Hartwig Medical Foundation Australia, Sydney, Australia. [6] Hartwig Medical Foundation, Science Park 408, 1098 XH Amsterdam, The Netherlands. *email: ecuppen@umcutrecht.nl

The use of 5-Fluorouracil (5-FU) as an anticancer agent became routine practice soon after its primary synthesis in 1957, and remains essential in many chemotherapeutic regimens today[1]. The fluoropyrimidines, especially 5-FU, capecitabine, tegafur, and cytarabine, are currently the third most commonly used anticancer drug in the treatment of solid cancers, including colorectal and breast cancers, and over two million patients are estimated to be treated with fluoropyrimidines each year[2]. Response rates of 5-FU as a single drug are 10–15%, but increase drastically (>50% response) when given in combination therapies with leucovorin together with oxaliplatin or irinotecan (i.e., FOLFOX and FOLFIRI, respectively)[3–5].

The antifolate property of fluoropyrimidines is thought to be the principal mechanism of action. Fluoropyrimidines are intracellularly converted into the antifolate 5-fluorodeoxyuridine monophosphate (5-FdUMP) that can form a covalent intermediate with the folate-dependent enzyme thymidylate synthase (TYMS)[6] Consequently, the formation of dTMP from dUMP is inhibited which results in an imbalance of the nucleotide pool that affects DNA synthesis, possibly through incorporation of uracil, and impairs genome replication, with negative consequences for rapidly dividing cells such as cancer cells. Moreover, it has been proposed that 5-fluorodeoxyuridine triphosphate (5-FdUTP) can be directly incorporated into genomic DNA as well[7,8]. Considering these properties, it is conceivable that fluoropyrimidines have mutagenic potential, although the mutational consequences of 5-FU treatments are still poorly understood.

In cancer, systematic analysis of genome-wide mutation catalogs has revealed a number of characteristic mutational patterns or mutational signatures[9]. Some of these signatures have been linked to perturbed endogenous processes like deficient DNA repair, or exogenous challenges, like exposure to UV-light or mutagenic chemicals. Such information thus provides insight into the mutational processes that have been active during tumorigenesis and which could potentially be used for prevention strategies or personalized treatment strategies. Previously, it has been shown that certain anticancer treatments can be associated with characteristic mutational signatures, such as alkylating agents[9,10], cisplatin[11,12] and ionizing radiation[13,14]. Unlike these anticancer treatments, and in spite of its mutagenic potential, 5-FU could thus far not be linked to any mutational signature using these systematic cancer cohort analyses.

Here, we assess the mutational consequences of fluoropyrimidines by exposing organoids of healthy intestinal stem cells to 5-FU followed by genome-wide analysis of single cells. For this, we use a previously described highly sensitive approach based on clonal expansion of individual cells followed by whole genome sequencing (WGS) for mutational spectrum analysis[15,16]. In vitro findings are subsequently validated by exploration of mutational patterns in breast and colorectal cancer patients who have had previous fluoropyrimidine treatments. Our results demonstrate that 5-FU induces both in vitro in organoids and in vivo in cancer cells a similar mutational pattern that is reminiscent of COSMIC signature 17.

## Results

**Characterization of 5-FU mutational effect in vitro.** We have set up human small intestinal (SI) isogenic organoid cultures which were exposed to 5-FU for 3 days followed by 4 days of recovery (Fig. 1a). This treatment procedure was repeated 5 times, which allowed the organoids to survive the exposure conditions and to accumulate a sufficient number of mutations. Then, individual organoid cells from the 5-FU exposed cultures were manually picked, expanded and analyzed by WGS with a read coverage-depth of ~30×. Somatic mutations were called against the original isogenic organoid line which was also sequenced at ~30×. Lastly, mutations which arose after the single-cell-step were filtered out based on low variant allele frequencies (Supplementary Fig. 1). A total of 1324 highly confident induced single base substitutions (SBSs) were identified in the autosomal genome that were accumulated during 5-FU treatment ($n = 2$ organoid lines). Organoids grown in parallel, but not exposed to 5-FU, served as control ($n = 6$ organoid lines). Not unexpectedly, untreated control organoids were found to proliferate faster than treated organoids, which makes it impossible to accurately determine the mutation accumulation load per cell division, although qualitative aspects and relative mutation contributions can still be interpreted.

To dissect active mutational processes, we analyzed the 96 mutational spectra of the obtained SBSs with trinucleotide context in more detail. We observed a distinct mutation profile for 5-FU exposed organoids when compared to the background in vitro mutation spectrum of untreated control SI organoids (Pearson correlation = 0.26; cosine sim = 0.57) (Fig. 1b). The most striking differences are the T > G mutations in a CTT trinucleotide context (further referred as C[T>G]T mutations) and, to a lesser extent, C[T>C]T and G[T>G]T mutations, which together account for more than half of the total mutation profile of 5-FU-treated organoids. This illustrates that 5-FU induces a characteristic mutational pattern in vitro that is driven by a mutational process that generates SBSs with a chance of ~35% being a CTT>CGT mutation.

**5-FU-induced mutational pattern in human cancer.** To assess if the observed 5-FU mutational consequences can also be detected in vivo in human cancer samples, we explored cancer whole-genome sequencing data from metastatic cancer patients (Hartwig Medical Foundation database) for which treatment data is also available[17]. 65% of colorectal ($n = 352$) and 36% of the breast ($n = 450$) cancer patients in this data set underwent 5-FU based treatment (i.e., 5-fluorouracil, fluoropyrimidine, capecitabine or tegafur—further referred to as 5-FU) at any time prior to biopsy and WGS. We performed an unbiased de novo mutational signature analysis using non-negative matrix factorization (NMF)[18] on both cohorts with inclusion of the 5-FU exposed organoid data. NMF identified sixteen mutational signatures which all showed high similarity with well-described signatures in human cancer (Fig. 2a, Supplementary Table 1) (http://cancer.sanger.ac.uk/cosmic/signatures)[19–21]. Interestingly, a signature that was highly similar to the 5-FU in vitro mutation spectrum was found in the set of the de novo extracted signatures (Pearson correlation = 0.98; cosine sim = 0.98) (Fig. 2a). This signature, further referred as "5-FU signature" (Fig. 2b), is predominated by C[T>G]T mutations (36%) which is almost equal to the 5-FU in vitro mutation spectrum (35% of C[T > G]T mutations). Ranking by the total mutational load of this 5-FU signature illustrates that patients who display a prominent contribution of this pattern were treated with 5-FU (Supplementary Fig. 2). These results indicate that 5-FU has the same mutagenic effect in vivo as in vitro.

**5-FU signature contribution in human cancer.** To quantify the mutational contribution of the 5-FU signature we compared 5-FU pretreated and non-5-FU pretreated patients (including a treatment-naive primary colorectal[22] and breast cancer cohort[23] as additional controls). The relative contribution of the 5-FU signature was calculated and compared for each patient to adjust for differences in tumor mutational burden (TMB—number of SBSs per Mbp) between primary and metastatic cohort[24]. In line

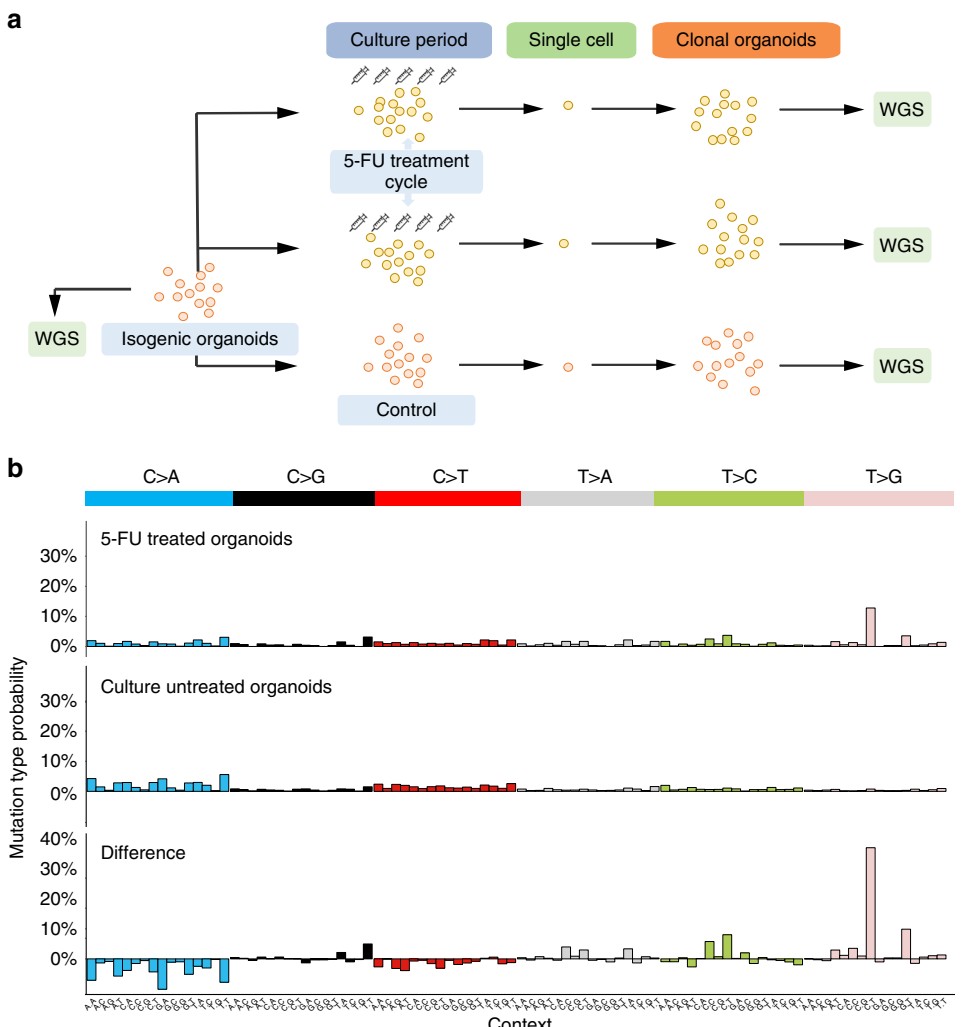

**Fig. 1** 5-FU induces context dependent T>G mutations in vitro. **a** Schematic overview of the experimental setup used to determine the 5-FU mutation spectrum in two independent human small intestinal organoid experiments. 6.25 µM 5-FU was added to isogenic organoids for 3 days, followed by a 4-day rest period. This cycle was repeated 5 times. Subsequently, organoids were made single cell and expanded further into clonal organoids to obtain sufficient DNA for WGS. Controls were cultured in 5-FU-free medium. The WGS data of the original isogenic organoid line served as reference sample. **b** The experimentally derived mutation spectra from 5-FU treated organoid lines (upper) and untreated organoid lines (middle). Each spectrum shows the mutation probability of each indicated context-dependent base substitution type. The spectrum below shows the difference between the 5-FU (positive values) and the in vitro (negative values) mutation spectrum

with our previous results, 5-FU pretreated patients showed a significantly higher 5-FU signature contribution compared to 5-FU untreated patients in both the colon and breast cancer cohort (both $P < 0.05$, Wilcoxon rank-sum test) (Fig. 2c). No significant differences were found between the 5-FU untreated patients and the treatment-naive cohorts. Examining the absolute mutational contribution for all extracted signatures shows that only the 5-FU signature is increased in contribution illustrating that 5-FU does not have a measurable impact on other signatures ($P < 0.05$, Wilcoxon rank-sum test, Supplementary Fig. 3). While 5-FU is most commonly used to treat breast and colon cancer patients, it is often also administered to patients with more rare cancer indications including pancreas ($n = 11$), biliary tract ($n = 6$) and head and neck ($n = 5$). In these cancer types, we identified the same 5-FU mutagenic effect as in breast and colon cancer, although not significant due to the low number of patients, which demonstrates that the 5-FU mutational process is tissue independent (Supplementary Fig. 4).

We observed an extensive variation in the number of 5-FU mutations per 5-FU treated patient ranging from 0 to roughly 15,000 mutations in both colon and breast cancer patients (Supplementary Fig. 2). This may be explained by variation in pharmacodynamics between patients, differences in the dosing and the duration of 5-FU treatment schedules[25], as well as by the evolution dynamics, but potentially also by other characteristics of the tumor. Indeed, analysis of tumor driver and suppressor genes ($n = 378$) uncovered that *TP53* mutated cancers accumulated more 5-FU mutations than *TP53* wild type cancers, both in colon and breast ($P < 0.05$, Wilcoxon rank-sum test, Fig. 2d). Also, fluorouracil and capecitabine were both found to be mutagenic in colon cancer, while in breast cancer only capecitabine showed an increased mutagenic effect (Supplementary Fig. 5), which might reflect differences between both tissues in drug uptake and treatment schemes. Notwithstanding the high variation in 5-FU signature contributions between patients, we observed that colon cancers overall have a higher 5-FU signature contribution than breast cancers, with a median mutation count of 1180 and 139 mutations, respectively.

The underlying clonal architecture of mutational events can be inferred from the variant allele frequency (VAF) and provides

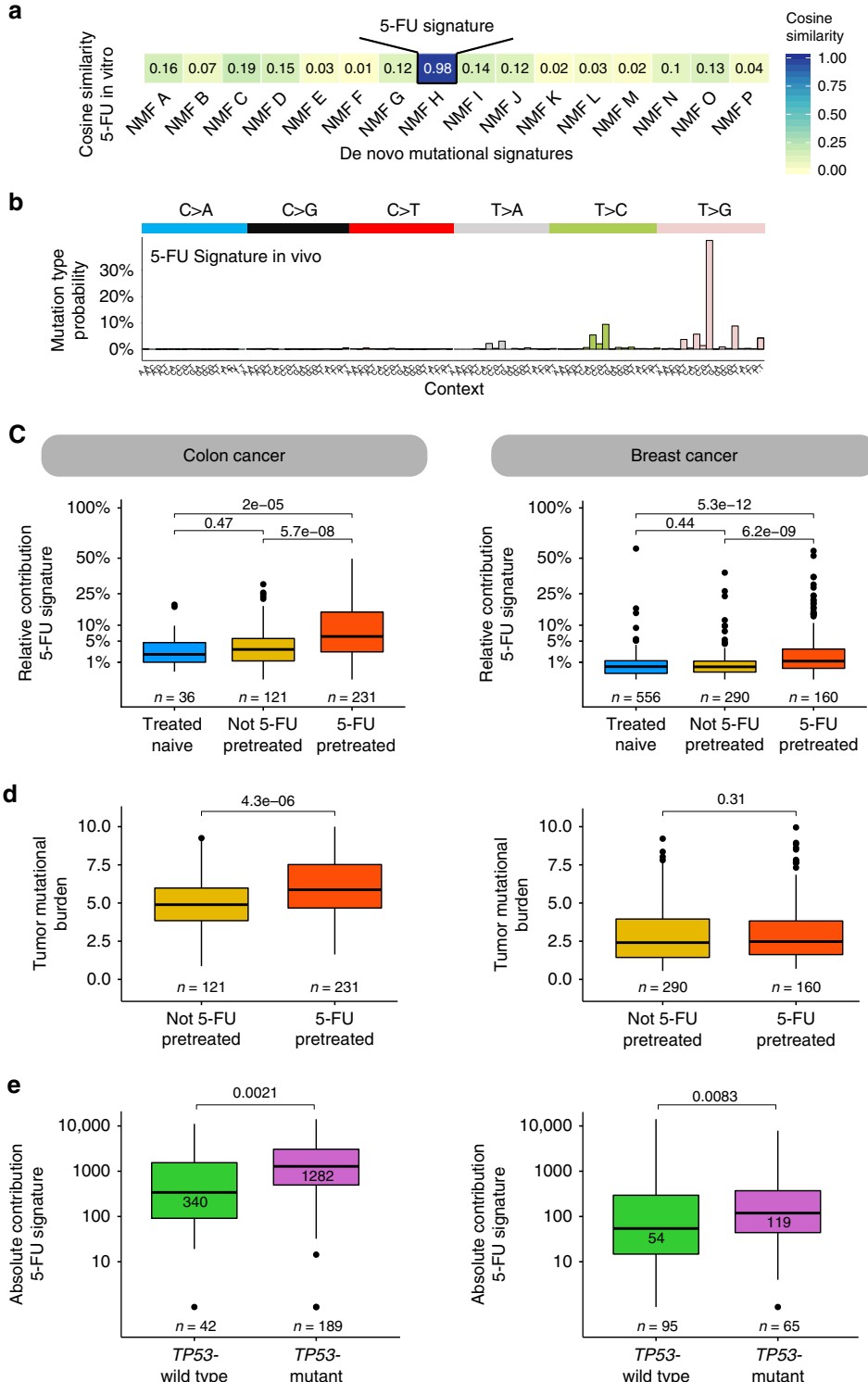

**Fig. 2** 5-FU mutational pattern and its contribution in human cancer. **a** Heatmap showing the cosine similarity scores for each de novo extracted signature with the in vitro experimental obtained 5-FU mutation spectrum. NMF H resembles the 5-FU experimental mutation spectrum (cos sim = 0.98) and is further assigned as the "5-FU signature" in the main text. **b** 5-FU mutation signature showing the mutation type probability for each context-dependent base substitution type. **c** Box-and whisker plots indicating the relative contribution of the 5-FU signature between 5-FU pretreated and not 5-FU pretreated colon (left) and breast (right) cancer patients with inclusion of the treatment naive cancer cohort. **d** Box-and whisker plots showing the tumor mutational burden (number of SBSs per Mbp) between 5-FU pretreated and not 5-FU pretreated cancer patients for the colon (left) and breast (right) cancer patients. **e** Box-and whisker plots showing the 5-FU mutational load between *TP53*-wild type and *TP53*-mutant cancers in 5-FU pretreated colon (left) and breast (right) patients. For all plots, a Wilcoxon rank-sum test between every cohort was performed and the *P*-value is illustrated at the top of the plots. All box- and whiskers plots display the first and the third quartiles (top and bottom of the box), the median (vertical line inside the box), the extremes (whiskers) and, if present, the outliers (single dots)

more insight into the timing of the activity of specific mutational processes. In comparison to clonal mutations, we found approximately a three-fold increase in the relative mutational contribution of the 5-FU signature for the subclonal mutations ($P < 0.05$, Wilcoxon rank-sum test, Supplementary Fig. 6). This points out that the 5-FU induced mutagenic activity is more profound in the metastatic colonies and therefore occurred at a later stage in tumor development, which is in line with the time of cancer diagnosis and subsequent 5-FU treatment.

**5-FU mutations in paired biopsies.** In the studied metastatic cancer patient cohort, 53 patients underwent two or more serial biopsies, which can be used to provide a more direct approach to study the chronological timing of the activity of mutational processes. This group of patients with multiple biopsies consisted of different cancer types of which 8 patients (colorectal cancer ($n = 4$) and breast cancer ($n = 4$)) that received a systemic 5-FU related treatment after the first biopsy and before one of the following biopsies. For every patient, we determined the mutation profiles of both biopsies and examined the difference in mutation numbers for each of the 96 mutation types, reasoning that 5-FU characteristic mutation types—particularly C[T>G]T mutations —would increase in mutational load. A mixed-effect regression analysis indeed revealed a positive correlation between the normalized absolute count of C[T>G]T mutations from the first biopsy compared to the second biopsy in patients treated with 5-FU (ANOVA linear mixed model; $P < 0.05$) (Fig. 3). Moreover, iterating this statistical analysis on each of the 96 possible mutation types resulted in significant $P$-values for all mutation types that are dominating the previously identified 5-FU signature (Fig. 3). Of note, no correlations were found between 5-FU characteristic mutation types and any other administered treatment drug (Carboplatin, Cisplatin, Oxaliplatin, Pazopanib, Pembrolizumab, and Pemetrexed) demonstrating that the signature is highly specifically induced by 5-FU (Supplementary Fig. 7).

**5-FU signature resembles COSMIC signature 17.** We compared the obtained 5-FU signature to the known COSMIC signatures and found a high similarity (Pearson correlation = 0.97; cosine

sim = 0.97) with COSMIC signature 17 (Fig. 4a), which is predominantly found in treatment-naive esophagus and gastric cancer. Recent work has split COSMIC signature 17 into two constituent signatures (SBS17a, predominantly characterized by T>C mutations and SBS17b, characterized by T>G mutations)[19], suggesting two distinct mutational processes. However, the here obtained 5-FU in vitro mutation spectrum showed both T>C and T>G mutations as in COSMIC signature 17, and thus our findings provide no evidence that COSMIC signature 17 exhibit a pattern of two independent mutational processes.

Next, we investigated whether the 5-FU signature also encompasses more detailed molecular features that are characteristic for COSMIC signature 17. In agreement with COSMIC signature 17[26,27], we also found a seven-base mutation context for C[T > G]T mutations in 5-FU pretreated colon and breast cancer patients which is predominated by A/T bases at the −4, −3 and −2 positions from the mutated base position (Fig. 4c). Furthermore, COSMIC signature 17 has been shown to display a higher mutation rate on the lagging strand[28,29]. Consistent with these reports, we observed a strong replication strand bias towards the lagging strand for C[T>G]T mutations types in 5-FU pretreated colon and breast cancer samples (Fig. 4b). In addition, we also noted a minor transcriptional strand bias in the colon samples for C[T>G]T mutations (Supplementary Fig. 8). Given this strong overlap in characteristics between both signatures, we conclude that the identified 5-FU signature is the same as COSMIC signature 17 and does not represent a novel signature.

**Impact on tumorigenesis.** We observed an average increase (~20%) in the overall TMB for 5-FU treated cancers, at least for the colon cancer patients ($P < 0.05$, Wilcoxon rank-sum test) (Fig. 2e). However, the 5-FU contribution on the TMB differs extensively per patient (Supplementary Fig. 9) where most 5-FU pretreated cancer patients (65% and 85% for colon and breast, respectively) show a limited impact of 5-FU on the TMB (<10%) and only a few patients (6% and 3% for colon and breast, respectively) demonstrate a substantial 5-FU contribution that affect the TMB with at least 30%. To investigate the impact of these 5-FU mutations on tumor evolution and disease progression, we selected all subclonal synonymous and non-synonymous

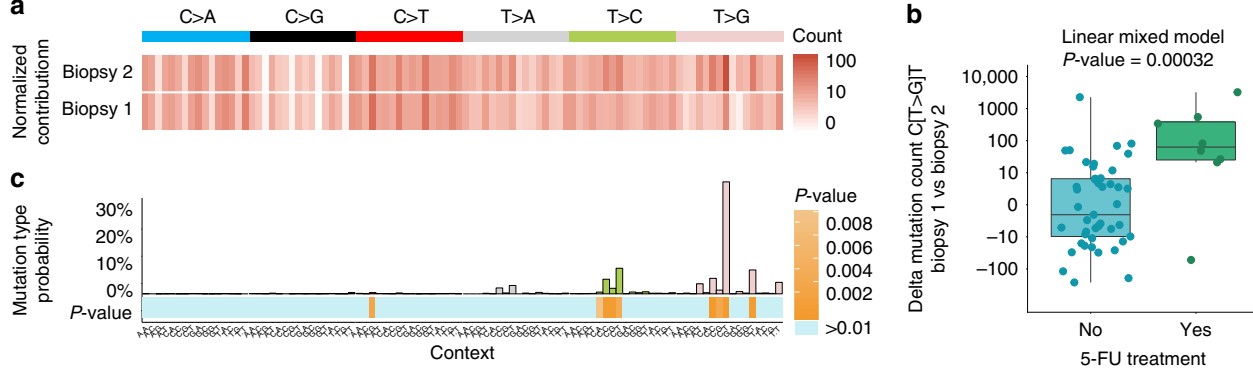

**Fig. 3** Mutational enrichment analysis for patients with multiple biopsies in 5-FU treated and 5-FU untreated patients. **a** Example heat map of one patient showing the normalized mutation count of every mutation type from the first (above) and second (below) biopsy. This normalization step was performed on both samples of each patient. **b** Linear mixed model regression analysis on the normalized mutation counts of one mutation type (here T[T>G]C mutations) between patients that received a 5-FU treatment between the two biopsies and patients not treated with 5-FU between two biopsies (see also Supplementary Fig. 7). In the model, we controlled for exposure dose and time as well as other therapies that were administered to the patient between the first and second biopsy. $P$-values were obtained by performing an ANOVA test on the regression model. Box-and whiskers plot displays the first and the third quartiles (top and bottom of the box), the median (vertical line inside the box), the extremes (whiskers) and the single data points (single dots). **c** Bar plot showing the mutation type probability for COSMIC signature 17 with below the obtained $P$-values from the linear mixed model for every mutation type. Note that most of the mutation types that characterize COSMIC signature 17 show a significant increase in normalized mutation count for patients treated with 5-FU between both biopsies

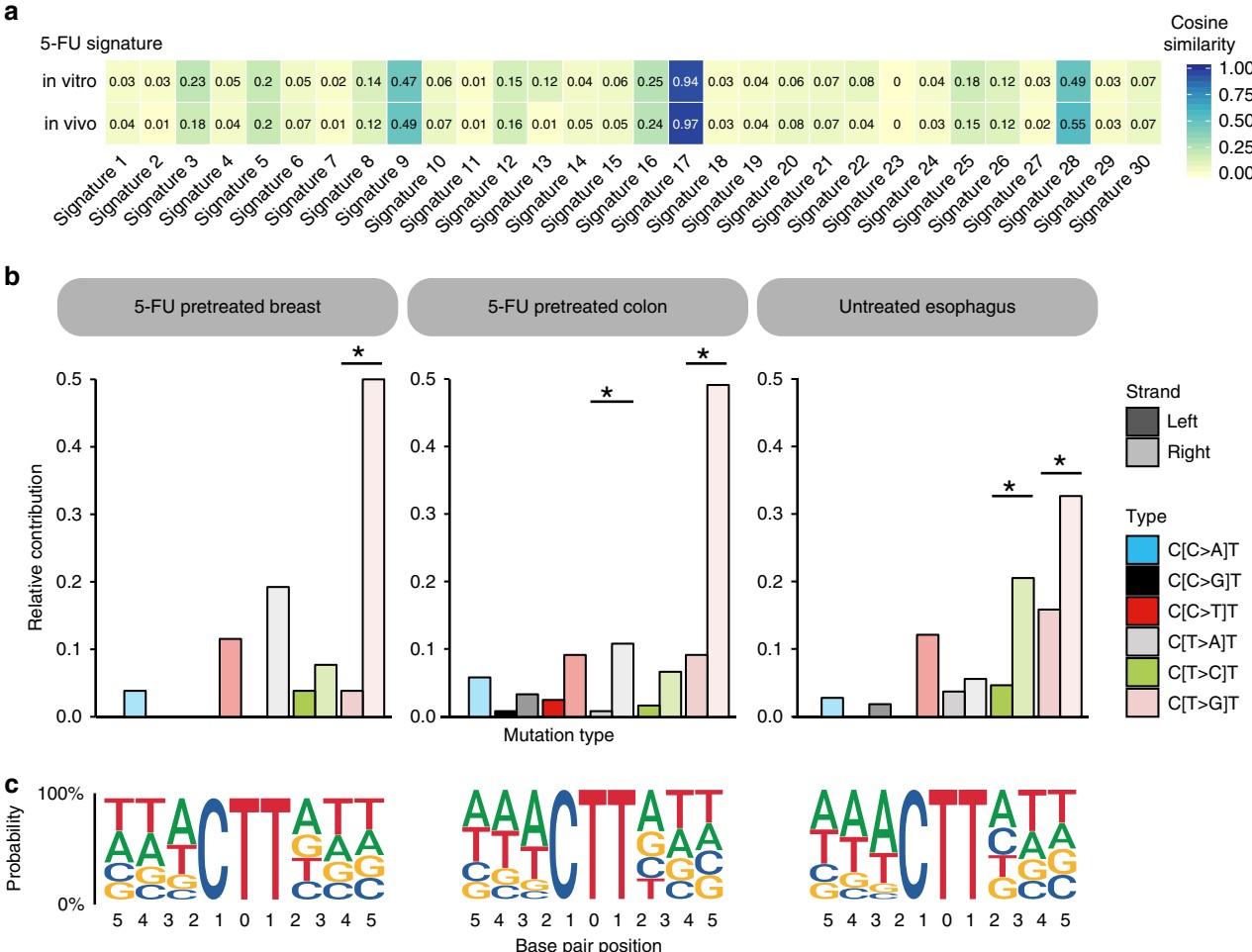

**Fig. 4** Comparison between the 5-FU signature and COSMIC Signature 17. **a** Heatmap showing the cosine similarity scores for the in vitro 5-FU mutation spectrum and the in vivo obtained 5-FU signature with the COSMIC signatures. Both patterns show a strong resemblance with COSMIC Signature 17. **b** Replication strand bias of C[N>N]T mutations in 5-FU pretreated colon and breast samples and not 5-FU pretreated esophagus samples. Relative levels of each base substitution type in the left (leading) and right (lagging) DNA strands are shown for each cohort. Asterisks indicate a significant difference ($P < 0.05$, two-sided Poisson test). **c** The eleven-base signature context of C[T>G]T mutations are presented as Logo plots. The mutated T is centered in each plot with fixed positions left (5′ direction) and right (3′ direction) from the mutation position

mutations that were most likely induced by 5-FU exposure for each patient (see Methods) to quantify oncogenic driver mutations induced by 5-FU (Supplementary Fig. 10). We observed no increase in the number of validated oncogenic drivers[30] in the 5-FU pretreated colon (5 driver mutations) and breast (5 driver mutations) cancer patients compared to non 5-FU pretreated colon (2 driver mutations, $P = 0.56$, Fisher exact test) and breast (5 driver mutations, $P = 0.26$, Fisher exact test) cancer patients (Supplementary Table 2).

In an attempt to characterize genes that may have contributed to 5-FU resistance, we performed a dN/dS analysis in which all single-nucleotide mutations and small insertions and deletions (INDELS) were included, but revealed no significantly mutated genes in contrast to resistance to hormonal therapies (e.g., ESR1 for breast and AR for prostate[17,31]) and targeted treatments (e.g., secondary BRAF mutations for melanoma treated with vemurafenib[32] and secondary EGFR mutations treated with EGFR inhibitors[33]).

Next, we investigated loss-of-function (LOF) and gain-of-function (GOF) events of key enzymes of the pyrimidine metabolic pathways. TYMS is considered as the key therapeutic target for 5-FU and overexpression of its gene has been linked to 5-FU resistance in in vitro as well as in in vivo experiments[34,35].

TYMS showed no LOF mutations in the breast and colorectal cohort, supporting the findings that TYMS is an essential gene[36]. On the other hand, GOF events of TYMS by means of copy number gains were found in 5-FU pretreated colon cancer patients ($n = 44$ out of 231) vs. untreated patients ($n = 8$ out of 121) ($P < 0.05$, Fisher exact test) (Supplementary Fig. 11), although this was not observed for breast cancer patients. This indicates a selective pressure towards increased levels of TYMS activity after 5-FU administration. The copy number level of TYMS seems to be inversely correlated with the absolute contribution of 5-FU pattern (Supplementary Fig. 11), which may suggest that TYMS overexpression can block the 5-FU mutational process by overcoming binding of 5-FdUMP by sheer number of TYMS protein.

It is interesting to note that, as we have shown with the organoid experiments, normal cells also accumulate 5-FU mutations. Consequently, it can be postulated that not only cancer cells, but any other cell in the body exposed to 5-FU may accumulate mutations that lead to the onset of secondary malignancies. To quantify this risk, we modeled the chance of introducing a cancer driver mutation resulting from 5-FU treatment, using the 5-FU specific mutation context and in vivo observed average mutation rate (Supplementary Fig. 12). This

model estimates that about 300 oncogenic mutations are introduced in vivo in $10^8$ colon stem cells per 5-FU treatment, which is 50-fold higher than under normal conditions as a result of in vivo mutational processes associated with aging. One full cycle of 5-FU treatment, therefore, reflects 'normal' mutation accumulation in colon stem cells of about 20 years[16]. As such, the consequences of 5-FU administration may be limited for patients with age above 60–70 years, but can be significant for cancer patients at a relatively young age (20–30 years old). Furthermore, patients carrying germline predisposition variants (e.g., APC mutation in FAP syndrome resulting in the development of tumors at a relatively young age) are at increased risk for acquiring a second hit and may be a contraindication for 5-FU treatment. We modeled this scenario as well and found a 20-fold increase in risk as compared to non-treated patients, which is equivalent to reducing the average age of onset for tumor development in FAP patients with 10 years.

## Discussion

Here, we demonstrate a causal relationship between 5-FU treatment and COSMIC signature 17, characterized by C[T > G]T base substitutions.

This finding differs from a previous study that did not find a measurable mutagenic effect of 5-FU exposure in cultured chicken lymphoblasts[37]. This discrepancy might be due to differences in experimental conditions (5-FU dosage, mutation detection) or the in vitro models used. Indeed, non-human cell lines are known to differ in DNA damage susceptibility[38], e.g. exposing aflatoxin to cell lines, mouse tumors and human tumors results in great diversity in mutation profiles[39]. Likewise, cisplatin signatures characterized with cell lines of different model organisms[12,37] do not recapitulate the cisplatin patterns recently found in human cancer[11,19].

Since 5-FU is structurally similar to thymidine and uracil nucleotides and has previously been shown to interfere with nucleotide biosynthesis and nucleotide pools[40–42], a mutagenic effect of 5-FU was anticipated. However, the strong resemblance with a previously described signature that was already linked to a different potentially underlying mechanism was surprising. COSMIC signature 17 is the hallmark signature of esophageal and gastric cancers and the presence of gastric refluxate has been suggested to be the responsible mutagen in these cancer types. High COSMIC signature 17 contributions are occasionally found in non-5-FU treated patients diagnosed with other cancer types as well[9,23]. For instance, a comprehensive study dissected the intratumor heterogeneity of three treatment naive colorectal tumors, of which one displayed extensive signature 17 contribution[43]. Thus, signature 17 reflects the consequences of a mutational process that can be instigated by multiple triggers including 5-FU exposure.

Recent work has proposed that COSMIC signature 17 reflects the mutagenic consequences of the presence of oxidized dGTP nucleotides in the nucleotide pool[29]. Indeed, a number of studies have reported that the presence of oxidized guanine nucleotides (8-oxo-dGTP) increases the T>G mutation rate[44,45]. Accordingly, inhibition of enzymes responsible for the removal of oxidized nucleotides, such as MTH1, MTH2, and NUDT5, have been shown to promote T>G mutations as well[46]. Also, the flanking sequence context of the dominant mutation type of Signature 17 mirrors the context of the dominant mutation type of Signature 18. This mutational process has been linked to direct oxidation of guanine located inside the DNA[47,48]. It is, therefore, tempting to speculate that the oxidation of dGTPs in the nucleotide pool underlies COSMIC Signature 17. As such, the presence of bile refluxate would be a plausible explanation for the elevated levels of 8-oxo-dGTP in esophagus cancer[49]. However, a recent study showed that bile refluxate alone does not generate 8-oxo-dGTPs, but that bile acid also requires an acidic environment to promote the production of 8-oxo-dGTP. This was only found in the epithelial cells of premalignant Barrett's esophageal cells, which gained transporters for bile acids, potentially clarifying why healthy esophageal cells do not show Signature 17 mutations[49–51]. Based on this, one could hypothesize that 5-FU exposure induces a similar oxidative stress environment in the cell that generates 8-oxo-dGTP thereby stimulating T>G mutations in a C[T>G]T context. In line with this, 5-FU treatment is less cytotoxic when combined with antioxidants[52] and ROS production is directly correlated with 5-FU treatment[53,54].

An alternative explanation of the underlying mutational process of COSMIC Signature 17 observed in 5-FU treated patients can be attributed to an imbalance of the nucleotide pool by TYMS inhibition, which is considered to be the major drug target of 5-FU. The 5-FU metabolite 5-FdUMP hampers the synthesis of dTMP which results in a depletion of dTTPs in the nucleotide pool[55,56] and impaired dTMP biosynthesis results in accelerated rates of genomic deoxyuridine triphosphate (dUTP) incorporation[57,58]. Next to dUTPs, also the 5-FU related byproduct 5-FdUTP can be incorporated during replication, which results in the accumulation of U:A and 5-FU:A base pairs[56]. These mutation types largely recapitulate Signature 17 and for this reason nucleotide imbalance by TYMS inhibition is a plausible cause for the here observed 5-FU mutations as well, although the strong similarity with the process active in esophageal cancer is not easily explained. In any case, further experimental follow-up will be required to dissect the underlying molecular mechanisms and to conclude whether one mutational mechanism is responsible for 5-FU specific mutation accumulation or that the 5-FU signature is the result of multiple mutational processes operating simultaneously on the genome (e.g., 8-oxo-dGTP, dUTPs, and 5-FdUTPs) that are accompanied by DNA repair mechanisms (e.g., uracil removal by uracil-DNA glycosylase [UDG]). Indeed, recent work revealed that the base excision DNA repair machinery selectively corrects Signature 17 mutations depending on its position around the nucleosome[59]. The involvement of DNA repair might also explain why tumors deficient in the p53 DNA damage checkpoint regulatory pathway accumulate more 5-FU mutations. Interestingly, breast tumors with high contribution of Signature 17 mutations were recently shown to have poor prognosis[60].

Nevertheless, we found that the mutation contribution of 5-FU administration does not have a great impact on the total tumor mutational burden and the driver landscape of the cancer in the majority of the patients. However, as the mechanisms driving 5-FU resistance remains largely to be elucidated, it cannot be excluded that induced mutations contribute to this process.

Furthermore, we calculated that young cancer survivors exhibit an increased risk for developing chemotherapy-related second malignancies as 5-FU can accelerate the rate of introducing novel oncogenic mutations in normal cells. Therefore treatment decision makers must be aware of the increased risk factors of 5-FU administration to cancer patients at a relatively young age[61,62].

Here, we have shown that the administration of fluoropyrimidines activates a mutational process that results in a highly characteristic mutational signature and as such, contributes to the mutational landscape of human (cancer) cells. Moreover, our results indicate that distinct triggers or processes can be at the origin of highly similar mutational signatures. Insights from this study could serve as a basis for future research to elucidate when and how these mutagenic agents converge on similar molecular mechanisms.

## Methods

**Patient cohort.** We selected patients of the CPCT-02 (NCT01855477) and DRUP (NCT02925234) clinical studies, which were approved by the medical ethical committees (METC) of the University Medical Center Utrecht and the Netherlands Cancer Institute, respectively. This national initiative consists of nearly 50 oncology centers from The Netherlands and aims to improve personalized cancer. To this end, Hartwig Medical Foundation sequences and characterizes the genomic landscape for a large number of patients. Furthermore, genomics data is integrated with clinical data which consists of primary tumor type, biopsy location, gender, pre-treatment type before biopsy, and treatment type after biopsy. A detailed description of the consortium and the whole patient cohort has been described in detail in Priestley et al.[17]. For this study, we selected cancers with primary tumor location in the breast, colon, and esophagus. Next, we also included all sample IDs, irrespective of the primary tumor location, which underwent at least 2 biopsies. Samples for which pretreatment was not documented (hasSystemicPreTreatment = NA) were excluded from this study. All used sample IDs in this study can be found in our GitHub repository (https://github.com/UMCUGenetics/5FU/blob/master/data/invivo/Used_Sample_IDs.txt).

**Organoid culturing.** A signed approval was obtained by the medical ethical committee UMC Utrecht (METC UMCU) for using the human small intestinal organoid line strain STE072 under STEM protocol (METC 10/402). These isogenic healthy human small intestinal organoids were cultured as described previously[15]. In short, organoids were grown on Complete Human Intestinal Organoid (CHIO) medium, supplemented with 30% Adv+++ (Advanced DMEM F12 [Thermofisher], supplemented with glutamax [1%, Thermofisher], hepes [10 mM, Thermofisher], penicillin/streptomycin [1%, Thermofisher]), in house produced Wnt (50%)[63] and R-spondin (20%)[63], B27 supplement (1×, Thermofisher), nicotinamide (10 mM Sigma), N-acetylcysteine (1.25 mM, Sigma), Primocin (0.1 mg/ml, Invivogen), A83-01 (0.5 µM, Tocris Bioscience), recombinant noggin (0.1 µg/ml, Peprotech), SB202190 (10 µM, Sigma) and hEGF (50 ng/ml, Peprotech). Organoids were embedded in matrigel and medium was refreshed every 2–3 days. A titration series was performed ranging from 0 to 100 uM 5-FU (0, 3.13, 6.25, 12.5, 25, 50, and 100 uM). The selected concentration of 6.25 uM was where roughly 50% of organoids grew out further after the 5 cycles of treatment. The selected concentration (i.e., 6.25 µM) is lower than often used in acute dosing experiments as these conditions were found to kill or senescence all cells. CHIO medium containing 6.25 uM 5-FU was added to the organoids 5 days post seeding, for a period of 3 days, after which the 5-FU-containing medium was refreshed with 5-FU-free CHIO medium for two consecutive days. The organoids were then left to rest for 2 days. This 7-day treatment cycle was repeated for 5 weeks after which the medium was changed to standard medium again and the organoids were left to rest for an additional day. The organoids were then dissociated into single cells by trypsinization and plated in a limited-dilution series. This was supplemented with CHIO medium containing ROCK inhibitor (10 µM, Abmole) and hES Cell Cloning & Recovery Supplement (1×, Tebu-Bio). Subsequently, individual clonal organoids were manually picked and expanded to gain enough material for WGS.

**DNA isolation and WGS of organoid lines.** Organoids were dissociated and DNA was isolated using the QiaSymphony DSP DNA mini kit (Qiagen, cat. No. 937236). Libraries were prepared using the Truseq DNA nano library prep kit (Illumina, cat. No. 20015964). Paired-end sequencing was performed (2 × 150 bp) on the generated libraries with 30x coverage using the Illumina HiSeq Xten at the Hartwig Medical Foundation.

**Somatic mutation calling.** Somatic mutation data of the CPCT and DRUP project were kindly shared by HMF on September 1, 2018. To exclude differences in accuracy and sensitivity from somatic calling workflows between in vivo and in vitro data, we pulled the HMF somatic mutation workflow from https://github.com/hartwigmedical/pipeline and installed the pipeline locally using GNU Guix with the recipe from https://github.com/UMCUGenetics/guix-additions. Full pipeline description is explained by Priestley et al.[17], and details and settings of all the tools can be found at their Github page. Briefly, sequence reads were mapped against human reference genome GRCh37 using Burrows-Wheeler Alignment (BWA-MEM) v0.7.5a[64]. Subsequently, somatic single base substitutions (SBSs) and small insertions and deletions (INDELS) were determined by providing the genotype and tumor (or organoid for in vitro analysis) sequencing data to Strelka v1.0.14[65] with adjustments as described elsewhere[17]. To obtain high-quality somatic mutations that can be attributed to 5-FU exposure in the organoid lines, we characterized the mutations that have accumulated between the sequential clonal expansion step. As such, we only considered somatic mutations with a variant allele frequency between 0.3 and 0.7, as mutations that fall outside this range were potentially induced in vitro after the clonal step.

**Mutational signature analysis.** De novo mutational signature extraction was performed using the NMF package (v0.21.0) with 100 iterations[18]. Non-negative matrix factorization (NMF) is an unsupervised approach that decompose high-dimensional datasets in a reduced number of meaningful patterns. For in vivo samples, we ran NMF on the colon and breast cancer cohort including the two

organoid lines exposed to 5-FU and six organoid lines that were cultured in identical medium for 140–146 days. In order to characterize the optimal number of patterns, we compared the cophenetic correlation coefficient over the range of possible signatures and assigned sixteen de novo signatures. This set of de novo extracted signatures were compared to the COSMIC cancer mutational signatures (http://cancer.sanger.ac.uk/cosmic/signatures), to the expanded list of mutational signatures[19], and signatures from other studies[20,21] using the cosine similarity from the Mutational Patterns R package as a measure of closeness[66]. We also used Mutational Patterns to determine the absolute contributions of each de novo obtained signature for the metastatic and primary cohorts. Briefly, a vector of 96 trinucleotide context counts for each sample was fitted using non-negative least squares regression to a 96 × n (where n is the number of signatures) matrix consisting of the trinucleotide context probabilities for each signature. The relative contribution of each signature was calculated by dividing the absolute counts by the total mutation count (i.e. tumor mutational burden) of the sample.

**Paired biopsies.** To test whether the number of 5-FU specific mutations was higher in the sample biopsied after 5-FU treatment than in the sample before the treatment, we first determined the 96-mutation count table for each sample. Next, we normalized the absolute mutation count for each set of paired samples per patient using the median ratio algorithm from the Deseq2 package[67]. Subsequently, we performed a linear mixed effect analysis using nlme R package[68] on each mutation type to assess the relationship between the normalized mutation count for each mutation type and treatment. We entered all the different treatment drugs into the model that were administered to at least 3 patients after biopsy one (5-FU, Carboplatin, Cisplatin, Oxaliplatin, Pazopanib, Pembrolizumab and Pemetrexed), and added random effects to correct for exposure time and dose for each treatment drug as well as the pharmacogenetics on patient level. We repeated this analysis using the relative mutation count of each mutation type.

**Ploidy and copy number analysis.** We used PURPLE[17] to obtain high quality somatic ploidy and copy number (CN) regions (https://github.com/hartwigmedical/hmftools/tree/master/purity-ploidy-estimator). Briefly, this tool combines B-allele frequency (BAF), read depth and structural variants to estimate the purity and CN profile of a tumor sample.

**Clonality.** The determination of the clonality of each mutation was adopted from Priestley et al.[17]. Briefly, the local ploidy level of each variant was calculated by multiplying the tumor adjusted variant allele score, obtained from PURPLE, with the local copy number level. All variants with a score above 1 are considered as clonal. Variants exhibiting a score lower than 1 were searched for a subclonal peak using a kernel density estimation using a kernel bandwidth of 0.05 after plotting the variant ploidy scores of all variants of a sample. All variants present in the peaks below the peak of ploidy = 1 were considered as subclonal mutations. Samples having at least 500 subclonal mutations and show an overall 5-FU signature contribution (at least 5%) were included for the subclonal analysis.

**Estimation of tumor mutational burden.** The mutation rate per megabase (Mb) of genomic DNA was calculated as the total genome-wide amount of SBSs divided over the total amount of mappable nucleotides (ACTG) in the human reference genome (hg19) FASTA sequence file:

$$TMB = \frac{(SBS_g)}{\left(\frac{2858674662}{10^6}\right)} \tag{1}$$

In this study, we excluded hypermutant samples (>10 mutations/Mbp), as determined by Campbell et al.[69], as hypermutant samples have an impact on both absolute and relative mutation contribution analysis.

**Detection of significantly mutated genes.** Using all SBS and INDEL variants from protein-coding genes, we ran dNdScv[51] to find significantly mutated genes using all SBSs and INDELs variants from protein-coding genes. This model can test the normalized ratio of each non-synonymous mutation type individually (missense, nonsense, and splicing) over background (synonymous) mutations whilst correcting for sequence composition and mutational signatures. A global q-value ≤0.1 was used to identify statistically significant driver genes. A post hoc Fisher's exact test was performed to evaluate whether the number of mutations of individual genes were enriched between two cohorts.

**Transcription and replication strand bias.** To compare the replication and transcription strand bias between cohorts, we selected samples with a high COSMIC signature 17 contribution (absolute contribution >2000 mutations and relative contribution >25% (5-FU pretreated colon n = 41, 5-FU pretreated breast n = 9, not 5-FU pretreated esophagus n = 34). Next, we selected all the point mutations bearing a C[N>N]T context where N can be any nucleotide, reasoning that the majority of the C[T>G]T mutations can be attributed 5-FU exposure in colon and breast cancer and 5-FU independent mutational processes in esophagus cancer. Mutation types other than C[T>G]T can thus be considered as control.

To assess DNA replication strand, we downloaded replication sequencing (Replic-Seq) data from Tomkova et al.[29] who characterized the replication timing profiles from Haradhvala[70]. As in Tomkova et al, we used replication strand information of 1 Mbp regions near the left and right of each origin[29]. Next, we generated a mutation count matrix 12 (6 trinucleotides × 2 strands) for each sample with replication strand information using Mutational Patterns R package[66]. After counting the number of mutations on each strand per cancer type and mutation type, a Poisson test for strand asymmetry was performed to test for significance. Similarly, a mutation count matrix of 12 was generated containing transcription strand information of all point mutations with a C[N>N]T context that fall within a gene body. The transcribed units of all protein-coding genes are based on Ensembl v75 (hg19) including the introns and untranslated regions. After estimating the mutation rate on the transcribed and non-transcribed strands, also a Poisson test for strand asymmetry was performed to test for significance. This package contains also functions to determine the replication timing. In brief, all point mutations were checked whether these were located in an intermediate, early or late replicating region. Enrichment or depletion analysis of point mutations in these genomic regions was performed using genomic distribution functions from Mutational Patterns R package[66].

**Association of point mutations with mutational patterns**. We estimated which mutational process was most likely at the origin of each point mutation as previously done in Letouzé et al.[28]. In doing so, we considered the mutation category (substitution type and trinucleotide context (TNC)) and the relative contribution of each mutational signature from each tumor sample. The likelihood of a point mutation, with a certain 96 trinucleotide context (TNC), induced by mutation signature X from a sample Y can be expressed as follows:

$$rel\,TNC_{Sample\,y}^{Sig\,x} = \frac{abs\,Sample^{Sig\,x} * rel\,TNC^{Sig\,x}}{\sum_{Sig} abs\,TNC_{sample\,y}^{Sig}} \quad (2)$$

Where $abs\,Sample^{Sig\,x}$ is the absolute mutation contribution of signature X for that sample; $rel\,TNC^{Sig\,x}$ is the mutation type probability for a given TNC of signature X divided by the sum of the mutation type probability for that TNC of all mutation signatures; and $\sum_{Sig} abs\,TNC_{sample\,y}^{Sig}$ is the sum of absolute mutation contribution of that TNC for every signature in sample Y. Overall, the sum of $rel\,TNC_{Sample\,y}^{Sig\,x}$ for every signature of one point mutation from one sample is equal to 1. Subsequently, the relative contribution of a mutational signature to all mutations from multiple samples was retrieved as the cumulative $rel\,TNC_{Sample\,y}^{Sig\,x}$ likelihoods of every mutation of the whole cohort. All mutations with a score of higher than 0.5 for a given signature were considered to be originated from that signature and were fed into dNdSCV for selection analysis.

**5-FU induced cancer driver mutation risk**. We used quantitative in vivo data and qualitative mutational characteristics to model the number of oncogenic mutations as a function of the number of cells, in the absence of negative selection. We applied the following formula:

$$M^{active}(N) = 0.015 \cdot dp \cdot N \cdot \mu \cdot \sum_{\substack{X \in \{C, T\} \\ Y \in \{A, C, G, T\} \\ i,j \in \{A, C, G, T\} \\ X \neq Y}} \left( P_{iXj \to Y} \cdot \frac{n_{iXj \to Y}}{L} \right) \quad (3)$$

where $M^{active}$ is number of mutations that activate driver genes, $dp$ is depletion in coding sequence (CDS), $\mu$ is the mutation rate, $N$ is number of cells, $P_{iXj>Y}$ is chance on $iXj > Y$ mutation based on the mutation spectrum, $n_{iXj>Y}$ is the number of positions where $iXj > Y$ mutation result in oncogene activation and $L$ is the length of CDS.

We used the following parameters: 1.5% of the genome is exon coding; Mutational depletion (likely due to repair) from the coding sequence is 0.3094464 (results obtained from Blokzijl et al.[16]). On average 2000 extra mutations with 5-FU signature per year accumulate in tumors due to 5-FU treatment (data based on this study) − 40 mutations accumulate per year in absence of 5-FU (normal in vivo mutation spectrum, 25% ~ signature 1 & 75% signature 5—results obtained from Blokzijl et al.[16]). Colon cancer originates in one of the $10^8$ colon stem cells[71]. Signature 17 mutation chance with inclusion of trinucleotide context (5-FU pretreated) and signature 1 (25%) + signature 5 (75%) for non 5-FU treated model; List of validated oncogenic mutations (exists of roughly 10,000 tumor suppressor and driver variants, obtained from Tamborero et al.[30]. Coding sequence length of small intestinal cells: 22563618 bp; The average duration of a 5-FU treatment regime is 24 weeks (12 cycles consisting of 2 weeks).

**Comparison with treated naive cancer cohorts**. The SBSs were called using Varscan 2.0 and post filtered with a QSS score above 30. Full description of this cohort can be found in Schütte et al.[22]. Both cohorts comprise of treatment naive cancer patients.

**Statistics**. Unless otherwise stated, we performed a Wilcoxon rank-sum test to compare continuous variables (for instance the relative or absolute contribution of mutational signatures vs. treated and not treated) and a Fisher's exact test was used to evaluate categorical data (treatment vs. the occurrence of a certain mutation). All statistical tests were two-sided and considered statistically significant when $P <$ 0.05. R version 3.4.4 was used for the statistical analyses.

**Reporting summary**. Further information on research design is available in the Nature Research Reporting Summary linked to this article.

## Data availability
WGS data and corresponding clinical data have been obtained from the Hartwig Medical Foundation and provided under data request number DR-047. Both WGS and clinical data is freely available for academic use from the Hartwig Medical Foundation through standardized procedures and request forms can be found at https://www.hartwigmedicalfoundation.nl. The human sequencing data of the 5-FU treated and control organoid lines have been deposited at the European Genome-phenome Archive (http://www.ebi.ac.uk/ega/) under accession numbers (EGAS00001003592 and (EGAS00001002955), respectively. For the primary breast cancer cohort, we used the publicly available somatic mutations from BASIS cohort (BRCA-EU dataset from https://dcc.icgc.org/) which were downloaded from the ICGC data portal on August 2, 2017. This cohort consists of 560 primary breast cancers and has previously been characterized in detail[23]. Somatic mutations of 41 primary colon cancer samples were kindly shared by Max-Planck-Institute with a signed agreement for data and sample transfer (http://www.oncotrack.eu). All the other data supporting the findings of this study are available within the article and its supplementary information files and from the corresponding author upon reasonable request. A reporting summary for this article is available as a Supplementary Information file.

## Code availability
All code and filtered vcf files from 5-FU treated organoid lines are freely available at https://github.com/UMCUGenetics/5FU.

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

## Acknowledgements

This publication and the underlying study have been made possible partly on the basis of the data that Hartwig Medical Foundation and the Center of Personalised Cancer Treatment (CPCT) have made available to the study. We also thank Sabine Middendorp for sharing the intestinal organoid line. In addition, we would also like to thank USEQ from UMCU for sequencing the organoid lines. Lastly, we are particularly grateful to all cancer patients enrolled within CPCT project for making their data available for fundamental cancer research. This work was financially supported by Oncode Institute and NWO zwaartekracht Cancer Genomics.nl program funding to E.C.

## Author contributions

S.C., E.K., E.C. and A.V.H designed the research. S.C., N.B. and E.K. carried out the wet lab experiments. B.V.d.R. and A.V.H. analyzed the data. R.J. and S.B. provided bioinformatic support. J.W.M.M., M-L.Y. and P.P. provided patient data. B.V.d.R. and A.V.H. analyzed the patient data. S.C., B.V.d.R. and A.V.H. developed the theoretical modeling. S.C., A.V.H. and E.C. wrote the paper. E.C. and A.V.H. supervised the study. All authors proofread, made comments and approved the paper.

## Competing interests

The authors declare no competing interests.
