## [Peer Review File · Nature Communications]

REVIEWERS' COMMENTS:

Reviewer #1 (Remarks to the Author):

The authors have fully addressed my comments.

The observation of higher mutation burden in TP53 mutant cancers is an interesting addition.

Reviewer #2 (Remarks to the Author):

The authors have addressed all my comments in a satisfactory manner. I recommend this manuscript for publication.

Reviewer #3 (Remarks to the Author):

The authors improved the paper during the revision, and I was convinced by the reviewer responses.

Reviewers' comments addressed by the authors

REVIEWERS' COMMENTS:

Reviewer #1 (Remarks to the Author):

The authors have fully addressed my comments.

The observation of higher mutation burden in TP53 mutant cancers is an interesting addition.

We agree that higher 5-FU related mutational accumulation in *TP53* null mutant tumors is interesting, and we thank you for pointing us in that direction.

Reviewer #2 (Remarks to the Author):

The authors have addressed all my comments in a satisfactory manner. I recommend this manuscript for publication.

We would like to thank the reviewer for all his or her constructive comments which improved the paper.

Reviewer #3 (Remarks to the Author):

The authors improved the paper during the revision, and I was convinced by the reviewer responses.

Thank you for your positive feedback and valuable comments that helped us to clarify certain passages in our manuscript.